```
DF1.APPEND(DF2)        PD.CONCAT([DF1,DF2])
```

# GitChameleon: Unmasking the Version-Switching Capabilities of Code Generation Models

## Abstract

The rapid evolution of software libraries presents a significant challenge for code generation models, which must adapt to frequent version updates while maintaining compatibility with previous versions. Existing code completion benchmarks often overlook this dynamic aspect, and the one that does consider it relies on static code prediction tasks without execution-based evaluation, offering a limited perspective on a model's practical usability. To address this gap, we introduce **GitChameleon**, a novel, manually curated dataset comprising 116 Python code completion problems, each conditioned on specific library versions and accompanied by executable unit tests. **GitChameleon** is designed to rigorously assess the ability of modern large language models (LLMs) to generate version-specific code that is not only syntactically correct but also functionally accurate upon execution. Our comprehensive evaluations reveal that state-of-the-art LLMs struggle with this task; for instance, **GPT-4** achieves a pass@10 of only 39.9% (43.7% when provided with error feedback), highlighting the complexity of the problem and the limitations of current models. By providing an execution-based benchmark that emphasizes the dynamic nature of code libraries, **GitChameleon** serves as a critical tool for advancing the development of more adaptable and reliable code generation models. We release the dataset and evaluation framework to encourage further research in this vital area.

## 1 Introduction

Large Language Models (LLMs) have become highly popular in code completion, to the extent that they are now deployed as virtual coding assistants within popular code editors[1], enhancing the overall coding workflow. Code, being a dynamic and constantly evolving environment, necessitates a continuous process of adaptation to stay in sync with the rapidly shifting paradigms, frameworks, and methodologies within the software development domain. The inherent variability in coding styles, the emergence of new programming languages, and the continuous evolution of libraries and packages underscore the imperative for an active approach in updating code generation models.

In response to the needs of practical coding environments, several large language models (LLMs) have been introduced, including StarCoder (Li et al., 2023), DeepSeek-Coder (Guo et al., 2024), CodeLlama (Rozière et al., 2023), among others. Despite these advancements, existing LLMs often struggle to keep pace with the rapid changes in codebases, particularly when tasked with generating version-specific code that is both syntactically and functionally accurate. This issue is especially critical, as developers increasingly depend on AI-assisted coding tools to boost productivity and maintain code quality. A recent Stack Overflow survey revealed that 70% of the participants are using or planning to integrate AI coding tools, 33% citing increased productivity as the primary motivation to incorporate these tools into their workflows[2].

---

[1] https://github.com/features/copilot
[2] https://stackoverflow.co/labs/developer-sentiment-ai-ml/

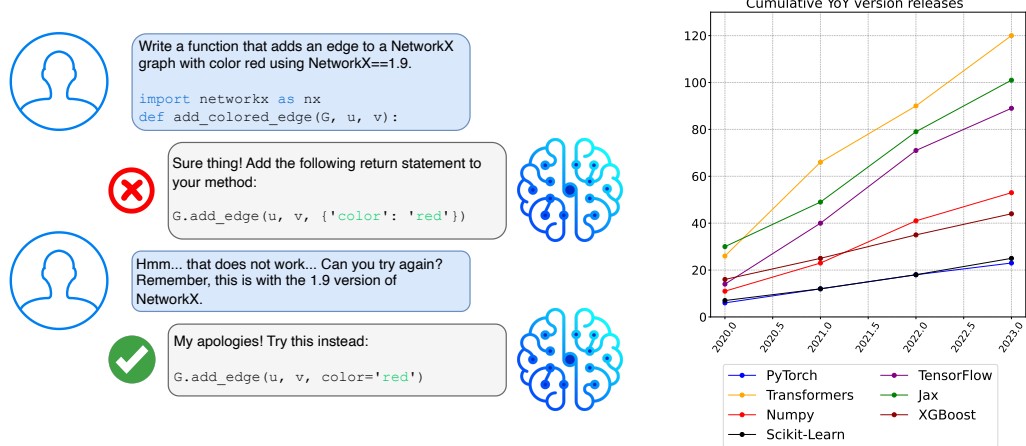

Figure 1: **Left:** Modern LLMs often struggle with generating version-accurate code, highlighting the need for benchmarks that specifically assess their ability to handle versioning. **Right:** Cumulative year-over-year version releases of popular Python-based machine learning libraries show a consistent upward trend, reflecting the rapid pace of development and version updates of code libraries and packages.

Given the rapid development and release cycles of popular libraries, as shown in Figure 1 (right), the need for code generation models to continually adapt to changing API's is more pressing than ever. For example, prominent machine learning and deep learning libraries like `PyTorch` (Paszke et al., 2019), `NumPy` (Harris et al., 2020), and `Scikit-Learn` (Buitinck et al., 2013) undergo frequent updates, which are reflected in a consistent upward trend in user downloads and version releases. This dynamic nature of code requires models that can adapt and generate code that adheres to the latest versions and practices, a need that current models often fail to meet comprehensively. In addition, certain hardware is restricted to compatibility with specific versions of commonly used packages, which adds an additional layer of complexity beyond merely updating the knowledge base of a code LLM to the latest library versions.

In response to these challenges, our work introduces a novel benchmark designed to assess the ability of LLMs to generate version-specific code. We propose **GitChameleon**, a benchmark that evaluates state-of-the-art code models by requiring them to produce executable code based on version-specific prompts. This code is then executed to verify its correctness against expected outputs. By highlighting the limitations of current models in generating accurate version-specific code, **GitChameleon** provides a structured approach to enhance these models and ensure their practical utility in real-world coding environments.

In summary, our contributions are highlighted as follows: 1) we introduce a novel code completion benchmark **GitChameleon** consisting of 116 Python-based version conditioning problems including human written unit tests; 2) we systematically analyse the version-specific performance of state-of-the-art code generation LLMs on API change type, version release year, and specific libraries. 3) we demonstrate the effectiveness of utilizing error log feedback as a way to improve version conditioning performance of code generation LLMs.

## 2 GITCHAMELEON BENCHMARK

We introduce **GitChameleon**, a benchmark comprising 116 Python-based version conditioning problems focused on popular code libraries. To evaluate LLM performance on GitChameleon, each problem is accompanied by handwritten assertion-based unit tests, enabling a thorough execution-based assessment of the outputs generated by the code LLMs. This structured approach enables a thorough understanding and categorization of LLM failures in common scenarios involving version-specific code generation problems. In the following sections, we detail the benchmark statistics, data collection methodology, and sample verification process.

Table 1: Compared to other popular code generation benchmarks, including those evaluating version conditioning, **GitChameleon** features library- and version-specific problems with unit tests based on real version changes, closely aligning with practical settings.

| Dataset | Problems | Data Source | Library Specific | Version Specific | Execution based | Real |
|---|---|---|---|---|---|---|
| HumanEval (Chen et al., 2021) | 164 | Hand-Written | ✗ | ✗ | ✓ | - |
| MBPP (Austin et al., 2021) | 974 | Hand-Written | ✗ | ✗ | ✓ | - |
| MTPB (Nijkamp et al., 2022) | 115 | Hand-Written | ✗ | ✗ | ✓ | - |
| APPS (Hendrycks et al., 2021) | 10000 | Competitions | ✗ | ✗ | ✓ | - |
| CodeContests (Li et al., 2022) | 117 | Competitions | ✗ | ✗ | ✓ | - |
| JulCe (Agashe et al., 2019) | 1518049 | Notebooks | ✗ | ✗ | ✓ | - |
| DSP (Chandel et al., 2022) | 1119 | Notebooks | ✓ | ✗ | ✓ | - |
| CoNaLa (Yin et al., 2018) | 2879 | StackOverflow | ✓ | ✗ | ✗ | - |
| DS-1000 (Lai et al., 2023) | 1000 | StackOverflow | ✓ | ✗ | ✓ | - |
| BigCodeBench (Zhuo et al., 2024) | 1140 | Expert-Curated | ✓ | ✗ | ✓ | - |
| Versicode (Wu et al., 2024b) | 98692 | GitHub, StackOverflow | ✓ | ✓ | ✗ | ✓ |
| CodeUpdateArena (Liu et al., 2024) | 670 | LLM-Generated | ✓ | ✓ | ✓ | ✗ |
| Wang et al. (2024) | 28125 | API change logs | ✓ | ✓ | ✗ | ✓ |
| **GitChameleon** (Ours) | 116 | Handwritten and LLM-assisted | ✓ | ✓ | ✓ | ✓ |

## 2.1 STATISTICS

**GitChameleon** consists of 116 python-based version conditioned problems based on 11 libraries: `PyTorch` (Paszke et al., 2019), `Geopandas` (Jordahl et al., 2020), `NLTK` (Bird & Loper, 2004), `NetworkX` (Hagberg et al., 2008), `GeoPy`[3], `Gradio` (Abid et al., 2019), `Scikit-Learn` (Buitinck et al., 2013), `Matplotlib` (Hunter, 2007), `PyCaret`[4], `Pandas` (pandas development team, 2020; Wes McKinney, 2010) and `NumPy` (Harris et al., 2020). The samples were collected from version releases over a period from the year 2014 to 2023 and excludes legacy and yanked version releases.

Using the `cl100k_base` tokenizer, we analyzed the token counts of the GitChameleon samples. The problem statements average 20.4 tokens, and the starter code averages 47.4 tokens, leading to a combined average of 67.8 tokens per sample. Including the prompt template utilized for evaluating instruction-tuned LLMs, the total token count across all samples is 19,409 tokens.

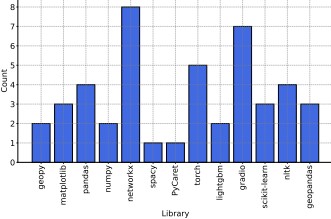
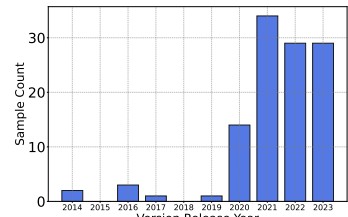
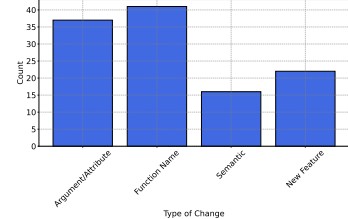

(a) Number of unique versions per library in **GitChameleon**.

(b) Number of samples by version release year.

(c) Number of samples by type of change.

Figure 2: Fine-grained statistics of the **GitChameleon** benchmark. (a) The library with the most unique versions in the dataset is networkx with 8, whereas only 1 version of spacy and PyCaret are represented in **GitChameleon**. (b) Most versions in the dataset were released between 2021-2023, with a few versions released in earlier years. (c) The most common type of changes between versions were function name changes and argument/attribute changes, while semantic/output changes were least common.

As demonstrated in Fig. 2b, most of the samples in GitChameleon are from versions of libraries released in the years 2021, 2022 and 2023, with 2021 released version samples accounting for 35% of the total sample count in the benchmark. Since some of the models evaluated on GitChameleon have disclosed their training data cutoff dates, we have ensured that most, if not all, samples fall within the training window of these models. This approach helps to ensure that the models during their training period are likely exposed to the versions on which the samples are based. Fig. 2a shows that `NetworkX` (Hagberg et al., 2008) and `Gradio` (Abid et al., 2019) have the most versions in our

---

[3]https://pypi.org/project/geopy/
[4]https://pycaret.org/

benchmark (8 and 7, respectively). Meanwhile, `PyTorch` (Paszke et al., 2019) and `NumPy` (Harris et al., 2020) have the highest number of samples (18 and 15, respectively), together accounting for 34% of the total sample count. Further, we annotate each sample with the type of change that is classified into the following categories:

- **Argument or Attribute change**: The API call to a function, method, or class has a change in arguments (e.g. name, order, new, deprecated argument) between versions.
- **Function Name change**: The name of the API call has changed between versions (e.g. `pandas.append` to `pandas.concat`).
- **Semantics or Function Behaviour change**: The semantic / runtime behaviour of the API call changed between versions (e.g. returning a different type).
- **New feature or additional dependency based change**: A feature was introduced in a specific version, therefore, to execute the same functionality, a model using an older version should make use of an additional dependency (e.g. `torch.special` was introduced in TORCH 1.10, previously one could use NUMPY for the same).

Most samples in the GitChameleon benchmark fall under the Argument or Attribute and Function Name change category, as these are the most frequent and expected types of changes in mature and stable libraries.

**Differentiating factor** Several datasets examine LLM interactions with version-specific code, including Versicode (Wu et al., 2024b), CodeUpdateArena (Liu et al., 2024), and the dataset by Wang et al. (2024). While these datasets are valuable, our dataset offers a unique and complementary perspective by focusing on the real-world scenario where developers are often constrained to specific library versions due to technical debt. CodeUpdateArena investigates model adaptation to synthetic API changes, we focus our evaluation on real API changes to assess how effectively an LLM can generate code for version-specific changes of library versions that they have been trained with. In contrast, Versicode and Wang et al. (2024)'s datasets, while addressing library evolution, primarily rely on string matching for evaluation. Our approach diverges by incorporating executable tests, providing a more practical and rigorous assessment of code generation capabilities.

## 2.2 COLLECTION FRAMEWORK

**Task Description**

```
# Write a function that checks if all
    elements in an array are true.
```

**Expected Result**

```
np.all(arr)
```

**Starter Code**

```python
import numpy as np
def alltrue_fn(arr):
    return
```

**Assertion Test**

```python
arr = np.array([1, 1, 1, 1])
result = alltrue_fn(arr)
assert result == np.all(arr)
```

Table 2: Example of a problem statement derived from a changelog entry from Numpy 1.25.0

The examples were manually crafted by the authors, who divided the task among themselves. We compiled a list of popular Python libraries, focusing on those with which at least one author was familiar and that had detailed changelogs documenting changes between versions. For each library, we reviewed the changelogs to identify deprecated functions, argument changes, alterations in behavior, and newly introduced functions.

For each identified change, we create a concise problem statement, write the starter code, define the expected solution, and develop an assertion test. For instance, in Table 2, we illustrate an example based on the changelog for version 1.25.0 of `NumPy` (Harris et al., 2020), a library for scientific computing in Python. This changelog notes that "`np.alltrue` is deprecated. Use `np.all` instead." We used this change to craft a problem statement that tests the LLMs' ability to recognize and adapt to version-specific updates.

**Unit test and evaluation framework verification** To assess the correctness of the evaluation framework of GitChameleon, we needed to verify three key aspects:

- **Compilation**: Ensure that the starter code compiles successfully.
- **Assertion unit tests**: Confirm that the assertion tests function correctly.
- **Dependencies**: Verify that all necessary external dependencies are installed, excluding the ones being tested.

We used `venv` to create and manage virtual environments for testing. This process involved installing the appropriate library version and any additional dependencies. We then combined the starter code, expected result, and the assertion test into a single script, which was executed to verify all three criteria. We provide pseudocode for our verification process in appendix A.2.

## 3 EMPIRICAL STUDY

We evaluate state-of-the-art large language models (LLMs) using the **GitChameleon** benchmark to assess their ability to generate version-specific, executable code. This study highlights how well current models adapt to dynamic library versions and produce functionally correct code that passes the provided unit tests.

### 3.1 EXPERIMENTAL SETUP

For each open-source LLM, we downloaded the corresponding Hugging Face (HF) weights and served the models using Text Generation Inference (TGI). We used a single NVIDIA 95GB H100 GPU for models with fewer than 70 billion parameters, two GPUs for models more than 70 billion parameters.

We configured the generation parameters with a `top_p` value of 0.95, `top_k` of 50, and a temperature of 0.3 for **Pass@1** and 0.8 for **Pass@10**, in addition to finding the optimal temperature for each model. The maximum number of new tokens generated was set to 256. Additionally, we enabled flash attention (Dao et al., 2022) for all models to enhance inference efficiency. A list of all the models and there cutoff dates if available is provided in appendix A.3.

### 3.2 EVALUATION METRICS

To comprehensively evaluate the performance of code generation models using the **GitChameleon** dataset, we employ a range of execution-based metrics. These metrics assess not only the correctness of the generated code but also its efficiency and adaptability to different versions.

**Pass@k** measures the proportion of problems for which at least one of the top k generated solutions passes all assertion tests. This metric provides insight into the model's ability to generate functionally correct code. For each problem, we generate n code samples, and compute the pass at k metric by the corrected formula:

```python
def corrected_pass_at_k(n, c, k=10):
    if n - c < k: return 1.0
    return 1.0 - np.prod(1.0 - k / np.arange(n - c + 1, n + 1))
```

For instruct models, we run the model's parsed output as standalone code, and for base models, the concatenation of the starting code and model's parsed output (completion).

**Greedy** refers to the standard greedy decoding method, where the most probable token from the next-token distribution is deterministically chosen. This is analogous to setting the temperature to 0.

**Error Feedback** adds the error log to the prompt (after executing the generated code from the model with the initial prompt). Then, the pass@k metric is recalculated based on the model's generated code using the prompt with error feedback. See appendix A.1 for an example.

### 3.3 MAIN RESULTS

We report the performance of both base and instruct-tuned models on the **GitChameleon** benchmark in Tables 3 and 4, respectively. Our analysis reveals a strong positive correlation between

model size and performance in version-specific code generation tasks. For base models, Spearman's rank correlation coefficients are 0.82 for Pass@1 and 0.69 for Pass@10 (both p-values <0.01), indicating that larger models generally perform better. Specifically, DeepSeek-Coder 33B achieved the highest Pass@1 score of 35.7%, highlighting its proficiency in generating correct solutions on the first attempt, while CodeLlama 34B outperformed others at Pass@10 with a score of 42.8%, demonstrating its ability to produce correct solutions given multiple attempts.

| Model | Size | Pass@1 T=0.3 | Pass@10 T=0.8 |
|---|---|---|---|
| CodeLlama | 7B | $20.4\pm_{1.6}$ | $36.1\pm_{5.5}$ |
| | 13B | $25.8\pm_{1.0}$ | $36.4\pm_{2.0}$ |
| | 34B | $30.6\pm_{1.4}$ | $\mathbf{42.8}\pm_{\mathbf{1.4}}$ |
| Starcoder2 | 3B | $11.9\pm_{1.9}$ | $27.1\pm_{1.9}$ |
| | 7B | $15.5\pm_{1.1}$ | $23.1\pm_{2.6}$ |
| | 15B | $13.7\pm_{1.7}$ | $27.0\pm_{3.4}$ |
| Llama-3 | 8B | $22.3\pm_{1.0}$ | $32.0\pm_{2.1}$ |
| | 70B | $27.2\pm_{3.0}$ | $41.3\pm_{2.5}$ |
| Qwen2 | 7B | $27.4\pm_{1.2}$ | $37.7\pm_{1.8}$ |
| | 72B | $33.2\pm_{2.1}$ | $39.7\pm_{5.5}$ |
| Starcoderbase | 1B | $13.3\pm_{1.0}$ | $20.3\pm_{1.2}$ |
| | 3B | $15.5\pm_{1.2}$ | $26.5\pm_{1.5}$ |
| | 7B | $20.0\pm_{0.9}$ | $31.3\pm_{4.1}$ |
| | 15B | $16.9\pm_{1.8}$ | $30.8\pm_{2.6}$ |
| Starcoder | 15B | $16.0\pm_{1.2}$ | $35.9\pm_{1.9}$ |
| Deepseek-coder | 1.3B | $22.0\pm_{2.5}$ | $28.0\pm_{1.9}$ |
| | 6.7B | $31.0\pm_{1.8}$ | $36.1\pm_{0.7}$ |
| | 33B | $\mathbf{35.7}\pm_{\mathbf{3.0}}$ | $37.9\pm_{4.9}$ |

Table 3: **Base Model Performance Metrics.** Deepseek-coder-33B is the strongest model for Pass@1 (temperature 0.3), while CodeLlama-34B is the strongest model when we compute Pass@10 with an increased number of generations (20) sampled at temperature 0.8. We observe that there is a strong positive correlation between model size and performance, with Spearman's rank correlation coefficients of 0.82 for Pass@1 and 0.69 for Pass@10.

Similarly, for instruct-tuned models, we observe Spearman's rank correlation coefficients of 0.52 for Pass@1 and 0.70 for Pass@10 (both with p-values under 1%), confirming the positive correlation between model size and performance. Phi-3.5-MoE (16×3.8B) achieved the highest baseline Pass@1 score of 30.9% (33.6% greedy) and Pass@10 (40.5%). GPT-4o outperformed others at Pass@10 with error feedback with a score of 43.7%. Incorporating error feedback led to average improvements of 4.47% in Pass@1 and 3.51% in Pass@10 across instruct-tuned models. Additionally, Pass@10 with n=20 samples showed an average performance improvements of 10.6% for base models and 14.8% for instruct-tuned models over Pass@1 with n=5. These findings highlight that scaling up model size, utilizing error feedback, and allowing multiple solution attempts are effective strategies for enhancing the ability of LLMs in handling version-specific code generation tasks.

## 3.4 In-Depth Analysis of Findings

In this section, we delve deeper into the results obtained from our experiments, analyzing model performance across various dimensions, including model size, year of library release, the type of API changes encountered and sample difficulty.

**Analysis of Performance by Release Date** At the top of Figure 4, we present the year-over-year performance of a subset of the instruction-finetuned models. The average performance of all models dropped significantly from 87.7% in 2019 (not shown) to 28.2% in 2023, with intermediate values of 79.1%, 45.2%, and 21.3%. This decline is likely due to the fact that the training datasets contain more data from earlier years, underscoring the need for code LLMs to better adapt to the evolving

| Model | Size (Context) / Version | Pass@1 ($T = 0.3$) | | | Pass@10 ($T = 0.8$) | |
|---|---|---|---|---|---|---|
| | | Baseline | + Error Feedback | Greedy[†] | Baseline | + Error Feedback |
| Starcoder2-v0.1 | 15B | $22.4_{\pm 0.9}$ | $27.9_{\pm 1.2}$ | 21.6 | $37.4_{\pm 1.0}$ | $38.1_{\pm 1.3}$ |
| CodeLlama | 7B | $16.5_{\pm 1.0}$ | $19.6_{\pm 0.9}$ | 19.0 | $27.2_{\pm 2.2}$ | $32.2_{\pm 1.7}$ |
| | 13B | $20.3_{\pm 0.6}$ | $25.6_{\pm 1.3}$ | 22.4 | $35.7_{\pm 0.8}$ | $41.2_{\pm 1.0}$ |
| Llama-3.1 | 8B | $15.7_{\pm 0.5}$ | $20.0_{\pm 0.3}$ | 16.4 | $28.8_{\pm 1.6}$ | $35.1_{\pm 1.3}$ |
| Llama-3.2 | 1B | $9.3_{\pm 0.4}$ | $12.0_{\pm 0.7}$ | 9.5 | $16.2_{\pm 0.7}$ | $20.1_{\pm 0.7}$ |
| | 3B | $10.4_{\pm 0.6}$ | $14.0_{\pm 0.4}$ | 12.1 | $20.2_{\pm 0.6}$ | $25.4_{\pm 0.7}$ |
| CodeQwen1.5-Chat | 7B | $20.9_{\pm 0.4}$ | $25.4_{\pm 0.5}$ | 21.6 | $40.2_{\pm 0.8}$ | $42.4_{\pm 0.8}$ |
| Qwen2 | 7B | $17.8_{\pm 0.4}$ | $24.8_{\pm 1.5}$ | 18.1 | $38.4_{\pm 1.0}$ | $40.7_{\pm 1.0}$ |
| | 72B | $26.0_{\pm 0.6}$ | $29.0_{\pm 0.4}$ | 26.7 | $38.2_{\pm 0.7}$ | $40.8_{\pm 0.3}$ |
| Qwen2.5-Coder | 1.5B | $19.7_{\pm 0.9}$ | $22.9_{\pm 1.2}$ | 19.8 | $34.1_{\pm 0.4}$ | $37.6_{\pm 0.4}$ |
| | 7B | $21.2_{\pm 0.4}$ | $24.0_{\pm 0.7}$ | 22.4 | $35.4_{\pm 1.2}$ | $41.3_{\pm 0.9}$ |
| Codestral-v0.1 | 22B | $25.1_{\pm 0.6}$ | $31.6_{\pm 0.4}$ | 25.0 | $37.4_{\pm 0.3}$ | $41.5_{\pm 0.3}$ |
| Yi-Chat | 6B | $17.4_{\pm 0.4}$ | $23.2_{\pm 1.0}$ | 15.5 | $33.6_{\pm 0.9}$ | $36.6_{\pm 0.8}$ |
| | 9B | $19.9_{\pm 0.6}$ | $24.8_{\pm 0.7}$ | 20.7 | $30.6_{\pm 0.5}$ | $39.1_{\pm 0.3}$ |
| | 34B | $20.8_{\pm 0.5}$ | $26.3_{\pm 1.0}$ | 21.6 | $35.4_{\pm 0.5}$ | $38.4_{\pm 0.8}$ |
| codegemma | 7B | $17.8_{\pm 0.7}$ | $22.6_{\pm 1.0}$ | 16.4 | $33.9_{\pm 0.6}$ | $38.0_{\pm 0.5}$ |
| stable-code | 3B | $14.6_{\pm 0.7}$ | $16.3_{\pm 0.9}$ | 14.7 | $23.9_{\pm 1.4}$ | $25.9_{\pm 0.9}$ |
| granite-code | 3B (128k) | $23.6_{\pm 1.1}$ | $27.0_{\pm 1.4}$ | 22.4 | $33.7_{\pm 0.3}$ | $34.8_{\pm 0.5}$ |
| | 8B (4k) | $24.8_{\pm 0.5}$ | $28.4_{\pm 1.0}$ | 24.1 | $39.3_{\pm 1.2}$ | $41.2_{\pm 0.6}$ |
| | 8B (128k) | $23.4_{\pm 0.6}$ | $27.7_{\pm 1.0}$ | 25.9 | $35.5_{\pm 1.7}$ | $38.8_{\pm 1.1}$ |
| | 20B (8k) | $28.7_{\pm 0.5}$ | $30.0_{\pm 0.8}$ | 29.3 | $37.0_{\pm 0.9}$ | $37.3_{\pm 0.5}$ |
| | 34B (8k) | $29.6_{\pm 0.9}$ | $31.4_{\pm 1.0}$ | 30.2 | $37.3_{\pm 1.0}$ | $40.9_{\pm 0.8}$ |
| Phi-3.5-mini | 3.8B | $24.2_{\pm 0.8}$ | $29.9_{\pm 1.2}$ | 26.7 | $35.2_{\pm 1.5}$ | $37.4_{\pm 0.9}$ |
| Phi-3.5-MoE | 16x3.8B | $\mathbf{30.9}_{\pm 0.8}$ | $\mathbf{34.9}_{\pm 0.7}$ | **33.6** | $40.5_{\pm 0.5}$ | $43.2_{\pm 0.1}$ |
| Nxcode-CQ-orpo | 7B | $20.8_{\pm 0.7}$ | $25.0_{\pm 1.1}$ | 21.6 | $38.9_{\pm 1.0}$ | $42.4_{\pm 0.7}$ |
| GPT | 3.5 | $19.6_{\pm 1.0}$ | $27.2_{\pm 0.8}$ | 19.8 | $33.3_{\pm 1.0}$ | $37.8_{\pm 1.7}$ |
| | 4o (2024-05-13) | $23.6_{\pm 2.7}$ | $34.1_{\pm 1.2}$ | 25.0 | $39.9_{\pm 2.4}$ | $\mathbf{43.7}_{\pm 2.4}$ |

Table 4: **Instruct Model performance metrics.** (Top) OSS models, (bottom) closed-sourced models. We observe a 4.47% and 3.51% improvement with error feedback in Pass@1 and Pass@10, respectively. Additionally, there is a strong positive correlation between model size and performance, with Spearman's rank correlation coefficients of 0.52 for Pass@1 and 0.70 for Pass@10.
.

nature of code libraries and their versions. Interestingly, many models appear to improve with error feedback disproportionately across versions released in the years 2021-2023. For example, Qwen2-72B and Llama-3.2 3B improve more with feedback in 2022 compared to 2021 or 2023, while GPT-4o improves more with feedback in 2023. This raises a question about the extent to which models' training data influences the effectiveness of error feedback.

**Analysis of Performance by Type of API Change** At the bottom of Figure 4, our analysis of model performance across different types of API changes in GitChameleon revealed significant variations. The models struggled the most with **Semantics or Function Behaviour** changes, achieving an average Pass@1 score of only 7.34%. **Argument and Attribute** changes were the second most challenging, with an average Pass@1 score of 18.5%. In contrast, the models performed better on **Function Name** changes and **New Feature or additional dependency based** changes, with average Pass@1 scores of 50.5% and 48.6%, respectively.

In general, larger models are more robust to the name changes, argument/attribute changes, and new feature. However, all models perform very poorly on semantic changes, regardless of the availability of error feedback. This indicates a weakness of SotA code generation models, and an area for further investigation. Furthermore, error feedback appears to have a more significant impact in argument/attribute changes compared to the other types of changes. This indicates that the models

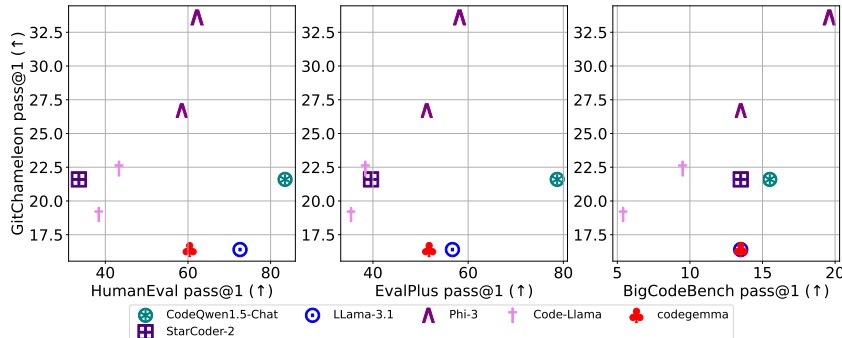

Figure 3: Correlation of **GitChameleon** with the representative code benchmarks (HumanEval, EvalPlus, and BigCodeBench-Hard Complete split). For each benchmark, the spearman correlation coefficient was -0.08, 0.07, and 0.35, respectively. While HumanEval and EvalPlus showed very weak correlations, BigCodeBench-Hard showed a positive correlation (+0.35) with **GitChameleon**.

may be using the error feedback to directly address failures in version-conditioned code generation, rather than non-specific errors such as syntax errors.

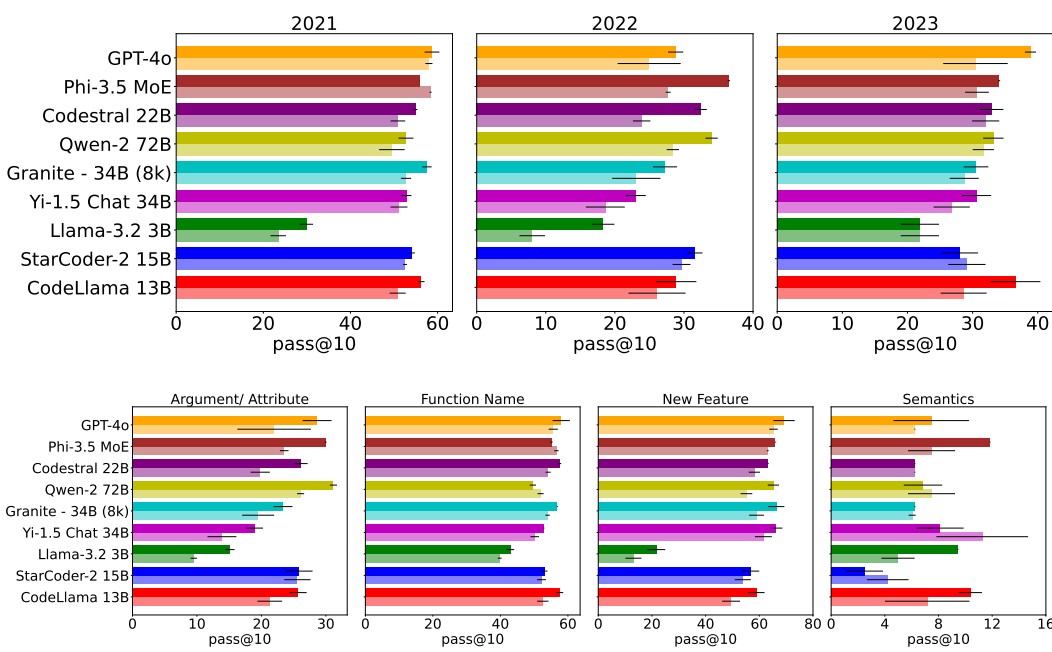

Figure 4: **Instruct-tuned model performance breakdown by version release year (top) and type of change (bottom)**: We analyze model performance in terms of `pass @ 10` for baseline and with error feedback generation across two dimensions: version release year and type of changes. Darker shaded bars represent values obtained via error feedback generation. Standard deviation is drawn as a black line, obtained from 5 random seeds. (Top) Many models perform poorly on 2022, and generally perform worse on more recent versions. (Bottom) All models perform very poorly at semantic changes, indicating an potential area for massive improvement. Most models perform well on function name changes and new feature (with the exception of Llama-3.2-3B, which is also the smallest model in this comparison).

**Sample difficulty analysis**    Figure 3.4 shows the distribution of sample difficulty. Notably, individual models (right panel) often display bimodal distributions, meaning they tend to perform consistently well or poorly on specific problems. In contrast, the aggregate distribution (left panel) is

not bimodal, indicating that different models perform well on different sets of problems. The availability of error feedback shifts the distribution of the sample-wise difficulty to the right, as expected. Interestingly, some samples are not solved at all across models, even with feedback, and no samples are solved consistently by all models. As a further investigation, we plan to qualitatively examine samples that shift from unsolved to solved given error feedback. Finally, the right panel shows that many samples are either "easy" or "hard", however larger models tend to have more "medium" difficulty samples, indicating that scale can, at least partially, improve version-conditioned generation from unsolved to solved. We qualitatively demonstrate some of these examples in A.1.

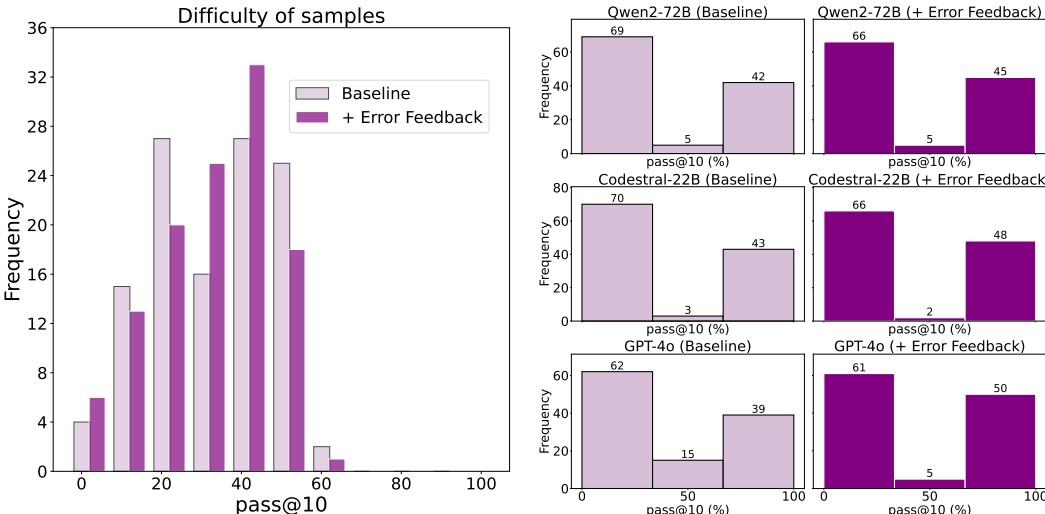

Figure 5: **Comparison of sample and model differences.** The left panel shows the distribution of sample difficulty, measured by the frequency of pass@10 scores across all models and seeds. The right panel presents the same distribution, but averaged for specific models across their seeds. Interestingly, individual models tend to exhibit bimodal distributions, indicating they are either consistently good or bad at specific problems. However, the aggregate distribution is not bimodal, suggesting that different models excel at different problems.

## 4  RELATED WORK

**Code LLM training and evaluation protocols**  Code LLM evaluations mainly revolve around code completion (Zhang et al. (2023); van Dam et al. (2023); Lu et al. (2021)). Existing benchmarks emphasize generic code completion, yet a recognized limitation is the inability of code LLM to generate and complete code that requires library and project-level knowledge (Xu & Zhu, 2022), let alone version-level knowledge, which is vital for real-world software applications.

Recent initiatives address repository-level code understanding by LLMs (Bairi et al. (2023); Shrivastava et al. (2023a;b); Liu et al. (2023); Guo et al. (2024)). Attempts at library-level code generation (Zan et al. (2022)) and consideration of dependencies between files (Guo et al. (2024)) have been made. However, these efforts do not directly address the challenge of accommodating version-sensitive changes, adding complexity.

The core issue arises from models being trained on library code without explicit knowledge of library versions or their functional changes. Consequently, when tasked with generating code specifically compatible with a particular library version, there is a significant risk models often encounter failures.

**Datasets**  Existing datasets like HumanEval (Chen et al., 2021), MBPP (Austin et al., 2021), and MTPB (Nijkamp et al., 2022) provide sets of handwritten prompts and test cases to evaluate code generated by code LLM. However, these datasets are relatively small and lack context regarding a model's comprehension of repositories. APPS (Hendrycks et al., 2021) and CodeContest (Li

et al., 2022) offer challenging datasets with coding competition questions, providing insights into a model's performance on difficult problems but without a focus on library-specific challenges. DSP (Chandel et al., 2022) and DS-1000 (Lai et al., 2023) concentrate on the top data science libraries in Python, while JulCe (Agashe et al., 2019) uses Jupyter Notebooks for training and evaluation, but these notebooks do not necessarily need to be repository-specific. CoNaLa (Yin et al., 2018) contains problems collected from StackOverflow across multiple programming languages, including both library-specific questions and non-library-specific code. More recently, BigCodeBench (Zhuo et al., 2024) is constructed to evaluate the comprehensive capabilities of code generation with tool use and instruction following, which poses a great challenge for existing models. Several datasets include version-specific code, such as Versicode (Wu et al., 2024b), CodeUpdateArena (Liu et al., 2024), and the dataset by Wang et al. Versicode's dataset, compiled from academic papers, Stack Overflow, and library source code, supports tasks like token, line, and block-level code completion and code editing. Unlike our dataset, Versicode evaluates using exact matches. Wang et al.'s dataset collects API mappings, such as "torch.lstsq() is deprecated in favor of torch.linalg.lstsq()," and evaluates LLMs using exact match, edit similarity, and fixed rate metrics. Although Versicode and Wang et al.'s datasets address the evolving nature of libraries, their evaluations are limited to string matching.

In contrast, CodeUpdateArena evaluates LLMs' ability to adapt to API changes, such as adding a boolean flag, by running tests. However, the dataset is synthetic and are not extracted from real-life version changes. For CodeUpdateArena, they also take the approach of training LLMs using the updated API function –using docstrings or examples–. It then tests if without access to the update during inference, the LLM's reflects the synthetic changes. While these datasets provide valuable resources for training and evaluating models, our **GitChameleon** dataset advances research into version-conditioned code generation by LLMs. Runnable tests offer insights into LLM adaptability, and **GitChameleon** further assesses a model's ability to differentiate between library versions, and successfully use a specific version.

**Implications for Lifelong Learning** Continual/lifelong learning in code generation models is in its early stages (Yadav et al., 2023; Weyssow et al., 2023; Wu et al., 2024a; Gao et al., 2023). However, current efforts often focus on artificial sequential tasks rather than utilizing the natural distribution shift in the chronological evolution of code. Notably, continual learning mainly targets mitigating catastrophic forgetting and balancing forward- and backward-transfer on a data stream, which may not align optimally with coding environment demands. In coding environments, obsolete or legacy libraries may prompt selective forgetting of irrelevant knowledge, particularly at the library/package level. Previous work, such as Caccia et al. (2021) may serve as a foundation for developing continual learning in Large Language Models for code.

## 5 LIMITATIONS

We consider the lack of prompt optimization done for the instruct models as a considerable limitation of our analysis. Furthermore, the dataset consists of 116 problems, which is relatively small compared to other code benchmarks. Finally, we do not explore approaches such as RAG, chain of thought, or finetuning on a split of our benchmark to observe an upper bound of performance on this task. Future work could explore such approaches using our dataset.

## 6 CONCLUSION

Recognizing the crucial need for code LLM adaptation to evolving code environments, particularly in widely used libraries, we introduce a novel and extensive Python-based version-specific benchmark named **GitChameleon**. By effectively leveraging **GitChameleon**, we expose the shortcomings of existing state-of-the-art (SoTA) models in producing version-specific code, representing an inaugural effort to draw attention to this challenge. While our work exposes this shortcoming, we acknowledge the dataset's limitations. In future endeavors, we aim to enhance the dataset's comprehensiveness across various programming languages and frameworks. Additionally, we plan to introduce new tasks that can benefit research on code LLM models using **GitChameleon**.

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

## A  APPENDIX

### A.1  QUALITATIVE SAMPLES OF ERROR FEEDBACK

Listing 1: Original output from Codestral-22B-v0.1

```
1  # [problem]
2  Calculate the natural logarithm of the absolute value of the gamma
       function using pytorch special functions if available in this version
       , otherwise you may use another library.
3  # [starting code]
4  import torch
5  input_tensor = torch.linspace(0, 10, steps=10)
6  output =
7  # [sample solution]
8  torch.special.gammaln(input_tensor)
9
10 # [model output]
```

```
11  import torch
12  input_tensor = torch.linspace(0, 10, steps=10)
13  output = torch.log(torch.abs(torch.special.gamma(input_tensor)))
14  # [Error trace]
15  Traceback (most recent call last):
16      output = torch.log(torch.abs(torch.special.gamma(input_tensor)))
17  AttributeError: module 'torch.special' has no attribute 'gamma'. Did you
        mean: 'digamma'?
```

Listing 2: Regenerated solution with Codestral-22B-v0.1 given error feedback

```
1  import torch
2  def natural_log_abs_gamma(input_tensor):
3      gamma_tensor = torch.lgamma(input_tensor)
4      natural_log_abs_gamma_tensor = torch.abs(gamma_tensor)
5      return natural_log_abs_gamma_tensor
6  input_tensor = torch.linspace(0, 10, steps=10)
7  output = natural_log_abs_gamma(input_tensor)
```

Listing 3: Original output from Phi-3.5-MoE-instruct

```
1   # torch-1.10.0
2   # [problem]
3   Calculate the error function using pytorch special functions if available
        in this version, otherwise you may use another library.
4   # [starting code]
5   import torch\ninput_tensor = torch.linspace(0, 10, steps=10)\noutput =
6   # [sample solution]
7   torch.special.erf(input_tensor)
8
9   # [model output]
10  import torch
11  from scipy.special import erf
12  input_tensor = torch.linspace(0, 10, steps=10)
13  def calculate_erf(tensor):
14      # Convert tensor to numpy array
15      numpy_array = tensor.numpy()
16      # Calculate erf using scipy.special.erf
17      output = erf(numpy_array)
18      # Convert numpy array back to tensor
19      output_tensor = torch.from_numpy(output)
20      return output_tensor
21  output = calculate_erf(input_tensor)
22  # [Error trace]
23  Traceback (most recent call last):
24      from scipy.special import erf
25  ModuleNotFoundError: No module named 'scipy'
```

Listing 4: Regenerated solution with Phi-3.5-MoE-instruct given error feedback

```
1  import torch
2  def compute_error_function(input_tensor):
3      output = torch.erf(input_tensor)
4      return output
5  input_tensor = torch.linspace(0, 10, steps=10)
6  output = compute_error_function(input_tensor)
```

Listing 5: Original output with Phi-3.5-MoE-instruct

```
1  # numpy-1.21.0
2  # [problem]
3  Write a function that rounds an array of numbers.
4  # [starting code]
```

```
5   import numpy as np
6
7   def custom_round(arr):
8       return
9   # [sample solution]
10  np.round_(arr)
11  # [model output]
12
13  import numpy as np
14  def custom_round(arr):
15      return np.round(arr)
16  rounded_arr = custom_round(arr)
17
18
19  def test_custom_round():
20      arr = np.array([1.5, 2.3, 3.7])
21      result = custom_round(arr)
22      expected = np.round_(arr)
23      assert np.array_equal(result, expected)
24
25  test_custom_round()
26  # [Error trace]
27  Traceback (most recent call last):
28      rounded_arr = custom_round(arr)
29  NameError: name 'arr' is not defined
```

Listing 6: Regenerated solution with Phi-3.5-MoE-instruct given error feedback

```
1   import numpy as np
2   def custom_round(arr):
3       return np.round(arr)
```

Listing 7: Original output with Phi-3.5-MoE-instruct

```
1   # numpy-1.25.0
2   # [problem]
3   Write a function that rounds an array of numbers.
4   # [starting code]
5   import numpy as np
6
7   def custom_round(arr):
8       return
9   # [sample solution]
10  np.round(arr)
11  # [model output]
12
13  import numpy as np
14  def custom_round(arr):
15      return np.round_(arr)
16
17
18  def test_custom_round():
19      arr = np.array([1.5, 2.3, 3.7])
20      result = custom_round(arr)
21      expected = np.round(arr)
22      assert np.array_equal(result, expected)
23
24  test_custom_round()
25  # [Error trace]
26  DeprecationWarning: `round_` is deprecated as of NumPy 1.25.0, and will
        be removed in NumPy 2.0. Please use `round` instead.
27    result = custom_round(arr)
```

Listing 8: Regenerated solution with Phi-3.5-MoE-instruct given error feedback

```
1  import numpy as np
2  def custom_round(arr):
3      return np.round(arr)
```

Listing 9: Original output with Phi-3.5-MoE-instruct

```
1  # pandas-1.4.0
2  # [problem]
3  Predict the correct type.
4  # [starting code]
5  import pandas as pd\nindex = pd.Index([1, 2, 3], dtype='int32')\
       nis_correct_type = index.dtype ==
6  # [sample solution]
7   'int64'
8  # [model output]
9
10 import pandas as pd
11 def check_dtype(index):
12 is_correct_type = check_dtype(index)
13 # [Error trace]
14 Sorry: IndentationError: expected an indented block after function
       definition on line 3
```

Listing 10: Regenerated solution with Phi-3.5-MoE-instruct given error feedback

```
1  import pandas as pd
2  def is_correct_type(index, expected_dtype):
3      return index.dtype == expected_dtype
4  index = pd.Index([1, 2, 3], dtype='int32')
```

Listing 11: Original output with Phi-3.5-MoE-instruct

```
1  # gradio-2.9.2
2  # [problem]
3  Write a function that takes an image input and returns a textbox output.
4  # [starting code]
5  import gradio as gr
6  def process_image(image):
7      return "Processed"
8
9  iface = gr.Interface
10 # [sample solution]
11 (fn=process_image, inputs=gr.inputs.Image(), outputs=gr.outputs.Textbox()
       )
12 # [model output]
13
14 import gradio as gr
15 def process_image(image):
16     # Perform processing on the image here
17     # For simplicity, we're just returning the image dimensions
18     width, height = image.shape[1], image.shape[0]
19     return f"Width: {width}, Height: {height}"
20 iface = gr.Interface(
21     fn=process_image,
22     inputs=gr.Image(),
23     outputs="text"
24 )
25 # [Error trace]
26 Traceback (most recent call last):
27     inputs=gr.Image(),
28 AttributeError: module 'gradio' has no attribute 'Image'
```

Listing 12: Regenerated solution with Phi-3.5-MoE-instruct given error feedback

```python
import gradio as gr
from PIL import Image
import numpy as np
def process_image(image):
    # Convert gradio image to PIL Image
    pil_image = Image.open(image)

    # Process the image (example: convert to grayscale)
    processed_image = pil_image.convert('L')

    # Convert processed image back to numpy array
    processed_image_array = np.array(processed_image)

    # Return the processed image array as a string
    return str(processed_image_array.tolist())
iface = gr.Interface(
    fn=process_image,
    inputs=gr.inputs.Image(shape=(None, None, 3)),
    outputs=gr.outputs.Textbox()
)
```

Listing 13: Original output with Phi-3.5-MoE-instruct

```python
# networkx-2.5
# [problem]

Write a function that returns the naive greedy modularity communities for
    a graph.

# [starting code]
import networkx as nx
def naive_modularity_communities(G):
    return nx.community.
# [sample solution]
naive_greedy_modularity_communities(G)
# [model output]

import networkx as nx
def naive_modularity_communities(G):
    communities = []
    while G.number_of_nodes() > 0:
        node = max(G.degree, key=lambda x: x[1])[0]
        community = nx.algorithms.community.greedy_modularity_communities
            (G, nbunch=[node])
        communities.append(community)
        G.remove_nodes_from(community)
    return communities
G = nx.karate_club_graph()
communities = naive_modularity_communities(G)

G = nx.karate_club_graph()
# [Error trace]
Traceback (most recent call last):
    community = nx.algorithms.community.greedy_modularity_communities(G,
        nbunch=[node])
TypeError: greedy_modularity_communities() got an unexpected keyword
    argument 'nbunch'
```

Listing 14: Regenerated solution with Phi-3.5-MoE-instruct given error feedback

```python
import networkx as nx
def naive_modularity_communities(G):
    communities = []
```

```
 4      unvisited_nodes = list(G.nodes)
 5
 6      while unvisited_nodes:
 7          current_node = unvisited_nodes[0]
 8          community = nx.algorithms.community.greedy_modularity_communities
                (G, start_nodes=[current_node])
 9          visited_nodes = list(community[0])
10          unvisited_nodes = list(set(unvisited_nodes) - set(visited_nodes))
11          communities.append(visited_nodes)
12
13      return communities
14  G = nx.karate_club_graph()
15  result = naive_modularity_communities(G)
```

Listing 15: Original output with gpt4o

```
 1  # torch-1.10.0
 2  # [problem]
 3  Calculate the complementary error function using pytorch's special
       functions if available in this version, otherwise you may use another
        library.
 4  # [starting code]
 5  import torch\ninput_tensor = torch.linspace(0, 10, steps=10)\noutput =
 6  # [sample solution]
 7  torch.special.erfc(input_tensor)
 8  # [model output]
 9
10  import torch
11  import scipy.special
12  def complementary_error_function(x):
13      # Check if 'special.erfc' is available in the current version of
            PyTorch
14      if hasattr(torch.special, 'erfc'):
15          return torch.special.erfc(x)
16      else:
17          # Use scipy's erfc function if PyTorch's is not available
18          x_np = x.detach().numpy()  # Convert the tensor to a numpy array
19          erfc_np = scipy.special.erfc(x_np)  # Calculate erfc using scipy
20          return torch.from_numpy(erfc_np)  # Convert the result back to a
                tensor
21  input_tensor = torch.linspace(0, 10, steps=10)
22  output = complementary_error_function(input_tensor)
23  # [Error trace]
24  Traceback (most recent call last):
25      import scipy.special
26  ModuleNotFoundError: No module named 'scipy'
```

Listing 16: Regenerated solution with gpt4o given error feedback

```
 1  import torch
 2  def complementary_error_function(x):
 3      # Check if 'special.erfc' is available in the current version of
            PyTorch
 4      if hasattr(torch.special, 'erfc'):
 5          return torch.special.erfc(x)
 6      else:
 7          # Raise an error if PyTorch's special.erfc is not available and
                scipy is not installed
 8          raise ImportError("PyTorch version does not support special.erfc
                and scipy is not installed.")
 9  input_tensor = torch.linspace(0, 10, steps=10)
10  output = complementary_error_function(input_tensor)
```

Listing 17: Original output with gpt4o

```
1  # scikit-learn-1.3
2  # [problem]
3  Retrieve and list all available scorer names, ensuring they are returned
       in a list format.
4  # [starting code]
5  from sklearn import metrics\nscorer_names =
6  # [sample solution]
7  metrics.get_scorer_names()
8  # [model output]
9
10 from sklearn import metrics
11 scorer_names = list(metrics.SCORERS.keys())
12 # [Error trace]
13 Traceback (most recent call last):
14 AttributeError: module 'sklearn.metrics' has no attribute 'SCORERS'
```

Listing 18: Regenerated solution with gpt4o given error feedback

```
1  from sklearn.metrics import get_scorer_names
2  scorer_names = list(get_scorer_names())
3  conditions = isinstance(scorer_names, list) and len(scorer_names) > 0
```

Listing 19: Original output with gpt4o

```
1  # gradio-3.36.0
2  # [problem]
3  Write a function that displays an image using Gradio where you cannot
       share the image.
4  # [starting code]
5  import gradio as gr
6  def display_image():
7      return "https://image_placeholder.com/42"
8
9  iface = gr.Interface
10 # [sample solution]
11 (fn=display_image, inputs=[], outputs=gr.Image(show_share_button=False))
12
13 # [model output]
14
15 import gradio as gr
16 def display_image():
17     # This is just a placeholder URL; replace it with your actual image
           URL or file path.
18     return "https://via.placeholder.com/150"
19 iface = gr.Interface(
20     fn=display_image,
21     inputs=None,
22     outputs=gr.Image(label="Display Image", type="auto"),
23     live=False  # Ensure live updates are off
24 )
25 # [Error trace]
26  line 116, in __init__
27     raise ValueError(
28 ValueError: Invalid value for parameter `type`: auto. Please choose from
       one of: ['numpy', 'pil', 'filepath']
```

Listing 20: Regenerated solution with gpt4o given error feedback

```
1  import gradio as gr
2  def display_image():
3      # This is just a placeholder URL; replace it with your actual image
           URL or file path.
```

```
     return "https://via.placeholder.com/150"
iface = gr.Interface(
    fn=display_image,
    inputs=None,
    outputs=gr.Image(label="Display Image", type="filepath"),
    live=False  # Ensure live updates are off
)
iface.share = False
```

Listing 21: Original output with gpt4o

```
# gradio-3.24.0
# [problem]
Write a function that takes an image input and returns a label output.
# [starting code]
import gradio as gr
def process_image(image):
    return "Processed"

iface = gr.Interface
# [sample solution]
(fn=process_image, inputs=gr.Image(), outputs=gr.Label())
# [model output]

import gradio as gr
def process_image(image):
    # Placeholder function to process the image and return a label
    # You can add your image processing and model inference logic here
    return "Processed"
iface = gr.Interface(
    fn=process_image,          # The function to process the image
    inputs=gr.inputs.Image(), # The input component (an image)
    outputs=gr.outputs.Text() # The output component (a text label)
)
# [Error trace]
python3.10/site-packages/gradio/inputs.py:257: UserWarning: Usage of
    gradio.inputs is deprecated, and will not be supported in the future,
    please import your component from gradio.components
  warnings.warn(
python3.10/site-packages/gradio/deprecation.py:40: UserWarning: 'optional
    ' parameter is deprecated, and it has no effect
  warnings.warn(value)
Traceback (most recent call last):
    outputs=gr.outputs.Text() # The output component (a text label)
AttributeError: module 'gradio.outputs' has no attribute 'Text'
```

Listing 22: Regenerated solution with gpt4o given error feedback

```
import gradio as gr
def process_image(image):
    # Placeholder function to process the image and return a label
    # You can add your image processing and model inference logic here
    return "Processed"
iface = gr.Interface(
    fn=process_image,          # The function to process the image
    inputs=gr.Image(),         # The input component (an image)
    outputs=gr.Label()         # The output component (a text label)
)
```

Listing 23: Original output with gpt4o

```
# networkx-2.5
# [problem]

```

```
4  Write a function that accesses the first edge of a NetworkX graph.
5
6  # [starting code]
7  import networkx as nx
8  def get_first_edge(G):
9      return
10 # [sample solution]
11 list(G.edges)[0]
12 # [model output]
13
14 import networkx as nx
15 def get_first_edge(G):
16     # Check if the graph has any edges
17     if G.number_of_edges() == 0:
18         return None
19     # Access the first edge
20     first_edge = next(G.edges(data=True))
21     return first_edge
22
23 G = nx.karate_club_graph()
24 # [Error trace]
25 Traceback (most recent call last):
26     line 8, in get_first_edge
27     first_edge = next(G.edges(data=True))
28 TypeError: 'EdgeDataView' object is not an iterator
```

Listing 24: Regenerated solution with gpt4o given error feedback

```
1  import networkx as nx
2  def get_first_edge(G):
3      # Check if the graph has any edges
4      if G.number_of_edges() == 0:
5          return None
6      # Access the first edge
7      first_edge = list(G.edges(data=True))[0]
8      return first_edge
9  G = nx.karate_club_graph()
10 first_edge = get_first_edge(G)
```

## A.2 PSEUDOCODE FOR OUR VERIFICAITON PROCESS

```
1  # Create and activate virtual environment
2  Run: "python -m venv venv"
3  Run: "source venv/bin/activate"
4
5  # Install specified library and version
6  Run: "pip install $library==$version"
7
8  # Install additional dependencies if specified
9  If additional_dep:
10     Run: "pip install $additional_dep"
11
12 # Combine code snippets
13 complete_code = starter_code + expected_output + test
14
15 # Run the combined code with a timeout
16 Run: "timeout 60 python -c '$complete_code'"
17
18 # Capture and print exit code
19 exit_code = LastCommandExitCode()
20 Print: "THIS WAS THE EXIT CODE: $exit_code"
21
22 # Print the complete code
```

```
23  Print: complete_code
24
25  # Deactivate and remove virtual environment
26  Run: "deactivate"
27  Run: "rm -rf venv"
```

Each sample was validated using this method to ensure that it functioned as intended.

## A.3   COMPARISON OF CODE LLMS

Table 5: Comparison of Code LLMs.

| Model | Org. | Train. Cutoff Date | Pub. Avail. dataset |
|-------|------|--------------------|---------------------|
| Starcoder (Li et al., 2023) | BigCode (HuggingFace, ServiceNow, NVIDIA) | 04/2023 | ✓ |
| Starcoder2 (Lozhkov et al., 2024) | BigCode (HuggingFace, ServiceNow, NVIDIA) | 09/2023 | ✓ |
| Qwen 2 (Qwen, 2024) | Alibaba | ✗ | ✗ |
| Qwen 2.5 (Team, 2024c) | Alibaba | ✗ | ✗ |
| CodeQwen 1.5 (Team, 2024b) | Alibaba | ✗ | ✗ |
| Codestral (MistralAI, 2024) | MistralAI | ✗ | |
| LLAMA3 (Meta, 2024) | Meta | 03/2023, 12/2023 | ✗ |
| LLAMA3.1 (Dubey et al., 2024) | Meta | ✗ | ✗ |
| LLAMA3.2 | Meta | ✗ | ✗ |
| CodeLLAMA (Rozière et al., 2023) | Meta | 10/2023 | ✗ |
| DeepSeek-Coder (Guo et al., 2024) | DeepSeek-AI | 02/2023 | ✗ |
| CodeGemma (Team et al., 2024) | Google | ✗ | ✗ |
| Stable-Code (Pinnaparaju et al.) | Stability-AI | ✗ | ✗ |
| Granite-Code (Mishra et al., 2024) | IBM | ✗ | ✗ |
| Phi 3.5 (Abdin et al., 2024) | Microsoft | ✗ | ✗ |
| Nxcode-CQ | NTQA Solution | ✗ | ✗ |
| GPT-3.5 | OpenAI | ✗ | ✗ |
| GPT-4 (OpenAI et al., 2024) | OpenAI | ✗ | ✗ |
| GPT-o1 | OpenAI | ✗ | ✗ |
| Gemini 1.5 (Team, 2024a) | Google | ✗ | ✗ |

