# OpenReview forum: "Unmasking the Version-Switching Capabilities of Code Generation Models"
_ICLR.cc/2025/Conference — ICLR 2025 Conference Withdrawn Submission_

### Official Review · Reviewer_igqt · 2024-11-03

**Soundness:** 2
**Presentation:** 3
**Contribution:** 2
**Rating:** 5
**Confidence:** 4

**Summary:**

The paper presents GitChameleon, a benchmark for evaluating large language models on version-specific code generation tasks. GitChameleon comprises 116 Python tasks tied to specific library versions, with unit tests for execution-based validation. The benchmark highlights current LLMs' limitations in handling dynamic library changes, offering a valuable tool for advancing version-aware code generation models.

**Strengths:**

This paper presents GitChameleon, a benchmark specifically designed to test the ability of LLMs in generating version-specific code. It includes 116 Python-based tasks tied to specific versions of popular libraries and equipped with executable unit tests. The paper presents an empirical study evaluating the performance of various state-of-the-art LLMs (e.g., GPT-4, CodeLlama, DeepSeek-Coder) on GitChameleon, highlighting their strengths and limitations in handling version-specific tasks. The paper is well-structured and clearly explains GitChameleon’s dataset creation, evaluation metrics, and model performance analysis, making it accessible to readers who may not be familiar with version-specific challenges in code generation.

**Weaknesses:**

While GitChameleon introduces a unique focus on version-specific code generation, the dataset is limited to 116 tasks and 11 Python libraries. This relatively small scale might restrict the generalizability of findings and the benchmark’s robustness. The evaluation lacks experimentation with widely used techniques in prompt engineering (such as chain of thought) and does not consider fine-tuning approaches like prompt-tuning or parameter-efficient tuning. GitChameleon’s current setup seems primarily focused on software engineering tasks and libraries. This specificity may make it more suitable for specialized conferences in software engineering (e.g., ICSE)

**Questions:**

1. Have you considered using additional techniques, such as prompt engineering, chain of thought (CoT), and prompt fine-tuning, to provide a more comprehensive evaluation of LLMs’ capabilities? Including these methods might better showcase the model's adaptability in version-specific code generation. We would like to see corresponding experimental results for these methods.

2. Could you elaborate on how this dataset could aid in improving the version-specific code generation abilities of existing LLMs? For instance, do you see potential in approaches like fine-tuning on version-specific tasks or using reinforcement learning from unit test feedback? We would be interested in seeing relevant experimental results for these approaches.

3. Would it be feasible to include more mainstream LLMs, such as Claude, in your evaluations? Additionally, do you plan to expand the dataset size? The current dataset is relatively small, which might limit the depth of the benchmark and its ability to generalize across various version-specific challenges.

---

### Official Review · Reviewer_QvUh · 2024-11-03

**Soundness:** 3
**Presentation:** 3
**Contribution:** 2
**Rating:** 3
**Confidence:** 4

**Summary:**

This paper proposes GitChameleon, an execution-based benchmark for evaluating LLMs’ coding capabilities in scenarios involving library updates. The benchmark is manually constructed with a joint effort from multiple authors. The paper provides a detailed analysis of the benchmark and an evaluation of multiple LLMs.

**Strengths:**

The paper studies an important problem in LLMs for code, i.e., dynamic updates of LLMs when it comes to library changes. I appreciate the authors’ effort in constructing GitChameleon and performing evaluation and analysis over it. This could be useful for future research.

**Weaknesses:**

- One major selling point of the paper is providing unit tests for assessing LLM-generated code. However, I don’t really get why unit tests are important for the domain of library changes. For the types of problems considered in this paper (i.e., Lines 167-176), the compiler or interpreter should already tell if a deprecated library version is used. Moreover, when the generated code passes the compiler or interpreter, do you ensure that the LLM actually uses the target API? The LLM could use some other features of the library to implement the same functionality. The LLM could also make errors on parts not concerning the target API.

- The dataset construction is a manual process. While I appreciate such an effort, this unfortunately results in manual bias and a relatively small size of data samples. For example, GitChameleon only covers 116 problems in Python libraries related to machine learning. This could threaten the validity of the results.

- The paper only provides an evaluation, without studying or discussing how to address the issue of LLMs in library updates. This is a relatively thin contribution, especially given that there are already a few benchmarks in the same domain as cited by the paper.

**Questions:**

I also have a few smaller questions:
- Line 177: How many samples fall under the Argument or Attribute and Function Name change categories exactly?
- Line 322: Why is performance better in 2023 than 2022?
- Line 430: Figure 3.4 should be Figure 5.
- Line 495: Wang et al. is not properly linked to the reference.

---

### Official Review · Reviewer_5kj9 · 2024-11-04

**Soundness:** 3
**Presentation:** 3
**Contribution:** 2
**Rating:** 3
**Confidence:** 4

**Summary:**

The paper introduces a new dataset and benchmark, GitChameleon, designed to evaluate the ability of large language models (LLMs) to handle version-specific code generation challenges in Python.
GitChameleon consists of 116 Python code completion problems, curated to test models' responses to specific library versions and accompanied by executable unit tests for functional validation.
This benchmark highlights the limitations of LLMs in managing code library version updates, which are critical for practical applications in dynamic software environments.

**Strengths:**

- The paper tackles the novel topic of assessing LLMs on version-specific code generation. Unlike existing benchmarks that generally focus on static code generation, GitChameleon is designed to evaluate models’ adaptability to changing library versions, a problem particularly relevant for real-world applications.

- The dataset is meticulously curated with executable unit tests, enhancing the robustness of its evaluations. GitChameleon addresses a critical gap in code generation benchmarking. By introducing an execution-based benchmark focused on version-specific compatibility, the work not only highlights current model limitations but also provides a concrete dataset for future improvements in LLM adaptability.

- The paper is well-organized and clear in its presentation.  The motivation behind GitChameleon is well presented as a crucial challenge in the field of code generation.

**Weaknesses:**

- Although the focus is clear and the dataset creation process is rigorous, the current GitChameleon dataset remains limited in size. Many libraries in the dataset include only a few API changes, which increases randomness and reduces the stability of the results.

- The benchmark’s limitation to Python makes the version-switching problem less complex in terms of **type information**. In other languages, such as Java, version updates often involve type changes, which are a significant aspect of the version-switching challenge. Thus, Python’s focus may oversimplify some key challenges in version-switching, limiting the paper’s broader contribution to this problem.

- Although the authors acknowledge methodological limitations, some baseline methods, such as RAG, should have been included as a critical evaluation aspect rather than overlooked. For practical benchmark usage, the authors should provide settings for RAG or few-shot learning, as these are more common approaches for addressing knowledge gaps in LLMs and should have been considered.

- Some related work in API learning, which is closely tied to the version-switching problem, is omitted. Including recent popular work on API learning could better contextualize the research. Relevant citations might include:

    - Zan, Daoguang, et al. "When language model meets private library." arXiv preprint arXiv:2210.17236 (2022).
    - Zhang, Kechi, et al. "Toolcoder: Teach code generation models to use API search tools." arXiv preprint arXiv:2305.04032 (2023).

In summary, the paper’s topic is interesting, but it appears to miss several critical dimensions of the version-switching problem, such as choosing a weakly-typed language and omitting a key aspect of version-switching. The current dataset may also be too limited to fully support the range of challenges outlined in the introduction section.

**Questions:**

In Section 2.1, the authors describe four types of changes. How do you ensure that this classification comprehensively captures the version-switching problem?

---

### Official Review · Reviewer_RyqN · 2024-11-11

**Soundness:** 2
**Presentation:** 3
**Contribution:** 1
**Rating:** 3
**Confidence:** 5

**Summary:**

The paper argues that LLMs are extensively used as virtual coding assistants but frequently fail to generate accurate version-specific code amidst the fast-evolving software landscape. To address this, the paper presents GitChameleon, a new benchmark that assesses and enhances LLMs' ability to produce executable, version-tailored code. This benchmark highlights existing models' limitations and offers a pathway to improve their effectiveness in dynamic coding environments.

**Strengths:**

- The paper underscores the need for LLMs to keep up with rapidly changing libraries, enhancing the practicality of code generation tools. Thus, GitChameleon focuses on version-specific code to evaluate model limitations and suggest improvements in LLM development.

- A new benchmark specifically targeting version-specific code generation.  The benchmark assesses models based on real execution of version-conditioned prompts, providing a systematic and practical measure of model performance.

**Weaknesses:**

- The benchmark assesses model performance on only a few hundred examples, which may not fully capture the diversity and complexity of real-world codebases.

- Expanding GitChameleon as the modes evolve will be non-trivial. The benchmark will soon be saturated. Also, to cover more libraries and languages may be difficult.

- While GitChameleon addresses an important gap, the benchmark and related evaluation seems to be an incremental improvement.

**Questions:**

How are you planning to keep the benchmark up-to-date as the APIs and the models will evolve?

---

### Note · Authors · 2024-11-20

I have read and agree with the venue's withdrawal policy on behalf of myself and my co-authors.